

# Research and Application of an Inner Thrust Measurement System for Rock and Soil Masses based On OFDR

Yimin Liu[1,2], Chenghu Wang[1], Pu Wang[1], Hao Zhou[1]

[1]Institute of Crustal Dyanmics, CEA, Beijing, 100085, China

[2]School of Manufacturing Science & Engineering, Sichuan University, Chengdu, 610065, China

*Correspondence to*: Yimin Liu (153973418@qq.com)

**Abstract.**

For measuring internal stresses in rock and soil masses, specifically on unstable slopes, the fiber Bragg grating (FBG) and optical time-domain reflectometer (OTDR) methods based on fiber sensing technology have disadvantages such as low

spatial resolution, low measurement accuracy, and non-distributed measurements. This paper presents a quasi-distributed thrust measurement system based on an optical frequency domain reflectometer (OFDR). First, the optical fiber stress sensing head was designed based on the micro-bending effect of the optical fiber, the cubic spline interpolation method was then used to compensate for the nonlinear effects of the OFDR stress sensing system, the compensation effects of different software methods were compared and analyzed simultaneously, which significantly improved the resolution and spatial-

positioning capabilities of the OFDR sensing system, and error calibration was then performed through laboratory experiments of lateral stress. The test results showed that the OFDR sensing system achieved a spatial resolution of 20 cm using a 500 m test fiber (the resolution of an OTDR sensing system is generally approximately 1 m), the maximum measurement pressure can reach 1.059 MPa and the maximum relative error is 8.9%. Finally, the field engineering application was carried out in the Chenjiagou landslide in Fengjie County, Chongqing City, Three Gorges Dam, China. The

application results showed that the system can accurately locate six fiber optic micro-bending stress sensors installed within the landslide body over a range of 0~420 m and can obtain the pressure values of their lateral thrusts. This system is a quasi-distributed stress monitoring instrument that provides long measurement distances, high spatial resolutions, high sensitivities, and fast responses that can be used for unstable slopes, slope engineering, water conservation and hydropower dams, and tunnel chambers, and thus, has good engineering application prospects in the safety monitoring field.

**1 Introduction**

For monitoring of unstable slopes, slope projects, water conservation hydropower dams, and tunnel chambers, surface displacement monitoring and underground deep displacement and stress monitoring methods are predominantly used domestically and abroad (Tang et al., 2012; Bellotti et al., 2014; Scaioni, 2015). GNNS and InSAR, close-range photogrammetry, and 3D laser scanning methods are surface monitoring methods used for unstable slopes (Gili et al., 2000;

Wright, 2004; Werner et al., 2007; Liu, 2014). These methods provide only basic data for the study of regional surface deformations of unstable slopes and the instability criteria for these methods are mostly based on displacement and deformation monitoring systems that cannot reflect the deformation and stress characteristics of deep rock and soil bodies. The deformation and development of a monitored body is a multi-dimensional and complex process. It is difficult to accurately detect the potential sliding position of an unstable slope and establish the corresponding relationships between the

time parameters, displacements and force changes. Therefore, early warning and prediction of the disaster body movement cannot be accurately achieved.

When increased deformation or a large local deformation of rock and soil occurs, inclinometer tubes deform sharply and drilling inclinometers cannot work using conventional monitoring methods. However, the sensing fiber in an optical fiber sensing system is able to work normally as long as it is not cut off; the sensing fiber contains many pressure-sensing points


40 in series. Moreover, the sensing fiber has strong resistance to scraping (Naruse et al., 2007). Among various methods, the optical time-domain reflectometer (OTDR), Brillouin time-domain reflection (BOTDR), and fiber Bragg grating (FBG) technologies have been applied for bridges, hydraulic engineering, construction, and geological hazard monitoring (Kee et al., 2000; Bao et al., 2002; Bernini et al., 2006). Kihara et al. placed optical fibers on the embankments of the Niyodo and Sendai Rivers in Japan and used polarized time-domain reflection to monitor the landslide displacements of the

45 embankments and achieved good results (Kihara et al., 2002). Dehua Liu et al. established a mechanical model of the FBG structure substrate-protective, layer-sensing fiber, analyzed the factors affecting the measurement accuracy of the FBG sensor, theoretically deduced the strain transfer relationship between the structural matrix and the fiber optic sensor under the action of tension and three-point pressure, obtained an analytical solution, and then proposed measures to improve the measurement accuracy (Liu et al., 2006). The University of Electronic Science and Technology in China has used the OTDR

50 technology to accomplish quasi-distributed monitoring of the inner thrusts of landslides and has conducted large-scale landslide monitoring in the Three Gorges Dam area (Zhang et al., 2006). Klar et al. used a distributed optical fiber component monitoring network to automatically detect disturbances, settlements, and other phenomena caused by tunnel excavation processes, and this optical fiber monitoring network provides a large amount of spatial distribution data and the tunnel mechanical model is analyzed and verified indoors using the measured optical fiber data (Klar et al., 2010). Surre et al.

55 placed distributed optical fibers on the surface of a steel bridge in the form of steel fiber ribbons; the sensing unit included a bare fiber sensor and a new bonded fiberglass tape with embedded fiber strain measurement capability and thermal compensation; the AQ8603 BOTDR strain analyzer was used to test the strain and stress distribution of the bridge during the loading process (Surre et al., 2013). Bin Shi et al. conducted monitoring analyses and health diagnoses of a tunnel using a BOTDR fiber strain gauge, proposed the installation of sensor fibers and temperature compensation methods, and discussed

60 the influence of environmental factors such as temperature and vibration on the measurement results (Shi et al., 2005). Minardo et al. used a BOTDR fiber strain gauge to perform static load tests on highway bridges and compared the data collected by fiber measurements with finite-element simulations and vibration-wire strain gauges to verify the effectiveness of the BOTDR method for monitoring large structural deformations (Minardo et al., 2011).

 At present, the main problems of the OTDR technology are lower spatial resolution, lower sensitivity, and lower

65 measurement accuracies. Compared with the OTDR, BOTDR technology can achieve simultaneous measurements of strain and temperature and has the advantages of high sensitivity, high measurement accuracy, and distributed measurements. However, its optical path structure is complex, signal adjustment is difficult, and the cost of the demodulator is expensive; therefore, the engineering practicality is poor and it is difficult to popularize. Compared with the OTDR and BOTDR, the optical frequency domain reflection (OFDR) presents the advantages of quasi-distributed monitoring, high spatial resolution,

70 high measurement accuracy, reliable performance, and strong anti-interference ability; therefore, the OFDR is suitable for internal stress monitoring of rock and soil masses during the entire process from creep to accelerated deformation.

 In this paper, the OFDR technology is used to measure the inner stresses of rock and soil masses. First, a fiber optic micro-bending stress sensor was designed as the stress detection device and a detuning filter algorithm was used to compensate for the spatial measurement errors created using the nonlinear sweep frequency band of the light source. In laboratory tests,

75 quasi-distributed stress measurements were realized within a sensing distance of 500 m. The spatial resolution was less than 20 cm, the maximum measurement pressure reached 1.059 MPa, and the maximum relative error did not exceed 8.9%. Field engineering application results show that the system can accurately sense stress locations and magnitudes.



## 2 Design the OFDR Sensing System

### 2.1 Design of the OFDR Sensor

Utilizing the characteristics of the bending loss in an optical fiber upon bending, the lateral stress can be converted into a physical quantity of optical power loss caused by the bending of the optical fiber for measurement (Takada, 1992; Liu et al., 2014). To convert the lateral stress into bending of the optical fiber, the sensor uses an elastic membrane as a pressure-sensitive element and a micro-bending modulation mechanism performs micro-bending processing on the single-mode fiber and senses the pressure distribution along the fiber axis (Zhou et al., 2004). As shown in Figure 1, the elastic membrane is

fixed on a rigid body and the periodic toothed pressure plate is composed of a moving tooth plate and fixed tooth plate. The moving tooth plate is fixed to the center of the elastic membrane and the fixed tooth plate is fixed to the base of the rigid body. The pressure (P) acts on the elastic membrane to generate the strain ($\varepsilon$). The moving tooth plate generates a corresponding displacement that changes the micro-bending amplitude of the optical fiber between the toothed plates, such that the loss due to micro-bending changes.

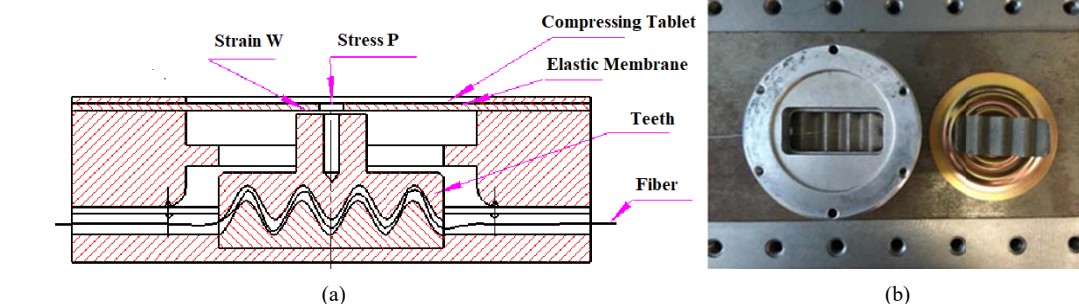


(a)                                        (b)

**Figure 1: Micro-bending pressure sensor (a) structural drawing; (b) physical image.**

When a weight is applied to the compressing tablet, the optical fiber in the sensor is bent, being squeezed by P, as shown in Figure 2. Therefore, it affects the light propagation in the optical fiber and the Rayleigh scattering signal, causing changes to

the transmission power. Figure 3 is a spectral diagram of a backward Rayleigh scattering signal after the sensor is subjected to stress, the solid line represents the case where the sensor is not subjected to stress, and the red line represents the case where the sensor is subjected to a 30 kg weight.

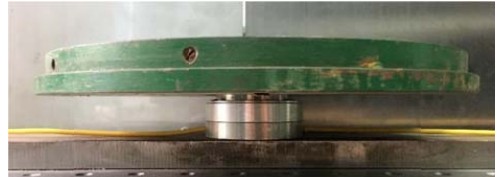

**Figure 2: Lateral stress acts on the sensor.**





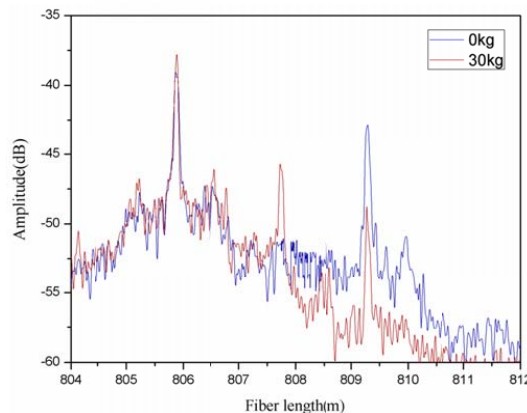


**Figure 3: Spectral diagram of a backward Rayleigh scattering signal.**

Figure 3 shows that for comparison with the case without stress, a reflection peak appears for the distance domain signal at 807.71 m when lateral stress is applied; this is because part of the light transmitted in the fiber is reflected back to the effects of fiber bending. Simultaneously, owing to the signal loss caused by the bending of the optical fiber, the amplitude of the

Rayleigh scattering signal after this position is reduced as a whole.

As shown in Figure 4, multiple OFDR micro-bending pressure sensors are arranged along the testing fiber, and the stress sensing array is connected in series. The OFDR demodulator can measure the stress distributions at multiple locations along the testing fiber.

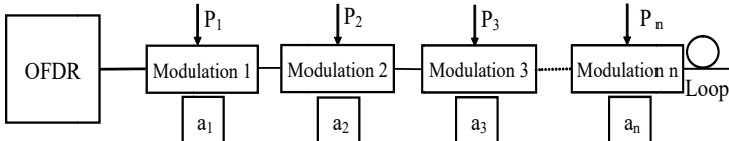

**Figure 4: Series connections of the OFDR sensing modules.**

**2.2 Composition of the Demodulator**

The system structural block diagram of the stress sensing demodulator based on OFDR is shown in Figure 5. A tunable laser was selected as the system frequency sweeping light source and its nonlinear effects were compensated by the software algorithms (Wu et al., 2017).

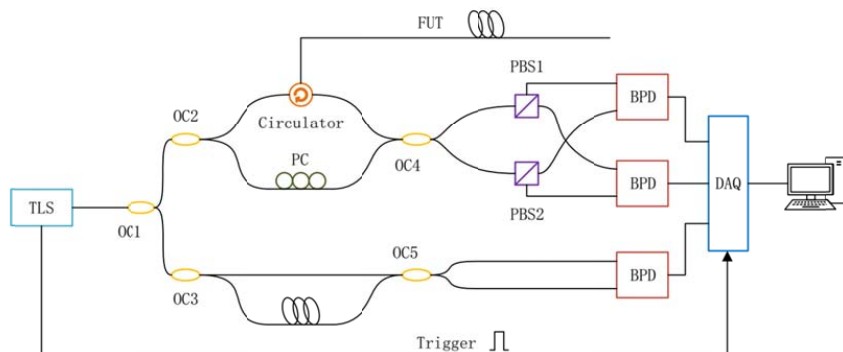


**Figure 5: Structure diagram of the OFDR demodulator (TLS: Tunable lasers, OC: Optical coupler, PC: Polarization controller, Circulator, PBS: Polarization beam splitter, BPD: Balanced detector, DAQ card, FUT: Testing fiber, and Trigger).**

The light source used in this demodulator is a Santec TSL-710 external cavity wavelength tunable laser. Its static line width is 100 kHz, the tunable range of the wavelength scanning speed is 0.5~100 nm/s, and the tunable wavelength ranges

1,480~1,640 nm. To compensate for the nonlinear frequency sweep effect of the light source, an auxiliary interferometer structure with a fixed delay is added to the main interferometer structure where the fiber under test is located; both the interferometer structures are MZ interferometer structures (Wang, 2015). To improve the signal-to-noise ratio of the coherent OFDR signals, the balanced detector used is a Thorlabs PDB430C with a bandwidth of 350 MHz; the acquisition card uses a Spectrum M4i.4421 with four channels and the highest sampling rate for each channel is 250 MHz. Additionally,

to suppress the polarization fading effect of the single-mode fiber, a polarization diversity receiving device has been added, the splitting ratio of the optical couplers OC1 is 95/5, OC2 is 99/1, and OC3, OC4, and OC5 are all 3 dB optical couplers, meaning that the splitting ratios are 50/50 for these couplers (Malatesta et al., 2000). To ensure that the wavelength scanning of the light source is synchronized with the data acquisition, the TTL trigger signal of the TSL-710 can be used as an external trigger signal for the data acquisition card. Since the transmission of light in the optical path takes time, the data

initially collected by the acquisition card are actually the data from the previous scan period, so the trigger delay needs to be set for the acquisition card. The trigger delay time is determined by the optical path delay time.

**3 Design of a Nonlinear Effect Compensation Algorithm**

To address the problem of low spatial resolution of the OFDR for long-distance measurements, this section uses the cubic spline interpolation method to perform accurate phase estimations for the light source outputs and a short-time Fourier

algorithm to obtain the time-frequency curve to determine the length of the delay fiber in the auxiliary interferometer. The nonlinear phase is then estimated by a high-order Taylor expansion to obtain the nonlinear phase of the intrinsic light.

**3.1 Cubic Spline Interpolation Method**

The cubic spline interpolation method is a type of one-dimensional interpolation method that is an algorithm which constructs a simple function on known discrete data and facilitates calculations of some unknown points in the interval. The

interpolation function curve constructed by this method is smoother but slower in terms of calculation speed (Song et al., 2012). As shown in Figure 6, the specific process for the cubic nonlinear spline interpolation for OFDR light source compensation is as follows:

(1) Obtain the discrete normalized instantaneous optical frequency information $v_1$, $v_2$, …, $v_n$ from the beat signal of the auxiliary interferometer. At this time, the intervals between $v_1$, $v_2$, …, $v_n$ are uneven.

(2) One-dimensional interpolation is performed on the original beat frequency data point $x(v_n)$ obtained by the main interferometer to get an interpolation curve; new uniform optical frequency interval points $v_1^{'}$, $v_2^{'}$, …, $v_n^{'}$ are then selected and the obtained interpolation curve is resampled to obtain a new data point $x(v_k^{'})$ for a uniform optical frequency interval. This process is shown in Figure 6.

(3) The new data point $x(v_k^{'})$ is transformed into the distance domain $X_G(z_n)$ by a standard FFT.

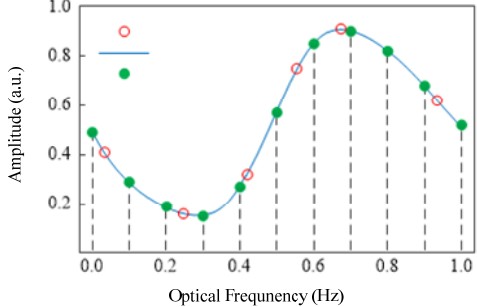




**Figure 6: Sampling diagram of the cubic spline interpolation (the red hollow circles are the primary data points of the main interferometer with a non-uniform optical frequency interval, the blue curve is a one-dimensional interpolation curve, and the green solid circles are the resampled points with a uniform optical frequency interval).**

### 3.2 Simulation Analysis

To verify the effectiveness of the cubic spline interpolation method in compensating for the nonlinear frequency sweep effect of the light source, this section uses LabVIEW software for simulation and then compares this simulation with the laboratory experimental data.

Assume that the nonlinear phase of the reference light is $e(t) = A_n \cdot \cos(2\pi f_n t)$, and $A_n$ and $f_n$ are the amplitude and frequency, respectively. The frequency sweep rate $\gamma(t)$ can be written as $\gamma(t) = \gamma_0 - (2\pi f_n)2 \cdot A_n \cdot \cos(2\pi f_n t)$, where $\gamma_0$ is the linear frequency

sweep rate and a constant term, $(2\pi f_n)2 \cdot A_n \cdot \cos(2\pi f_n t)$ changes with time as a sinusoid that is the interference term. To represent the degree of fluctuation of the frequency sweep rate $\gamma(t)$, the ratio of the amplitude $(2\pi f_n)2 \ A_n$ of the interference term of $\gamma(t)$ to the constant term $\gamma_0$ is defined as $K$. The larger the $K$ value, the greater the degree of fluctuation of $\gamma(t)$, indicating that the linearity of the frequency sweep of the light source is worse. In the simulation, the linear frequency sweep rate is taken as a fixed value $\gamma_0 = 625$ GHz/s and the corresponding wavelength sweep rate is 5 nm/s (the central wavelength

is 1,550 nm). Simultaneously, it is assumed that a strong reflection peak exists at a certain position on the optical fiber to be tested and the simulation function is shown in Equation 1.

$$I(t) = \cos\left\{2\pi\left\{\gamma_0\tau t + v_0\tau - \frac{1}{2}\gamma_0\tau^2 + A_n\cos\left(2\pi f_n t\right) - A_n\cos\left[2\pi f_n\left(t-\tau\right)\right]\right\}\right\} \qquad (1)$$

To facilitate the simulation calculations, the amplitude $I(t)$ is taken as 1. During the simulation, $I(t)$ needs to be discretized; the sampling rate is set to 1 MS/s and the sampling time is 1 s. Additionally, it is assumed that the position of the strong

reflection point along the optical fiber to be measured is at 20.55 m, and the corresponding group delay is $\tau = 2 \times 10^{-7}$ s. According to the magnitude of the nonlinear effect, it is discussed as follows:

(1) let $A_n = 5 \times 10^6$, $f_n = 25$ Hz and the group delay $\tau$ is still $6 \times 10^{-7}$ s, corresponding to the fiber position 61.65 m; at this time the $K$ value is still 0.2 and the other parameters are unchanged. The effect of the one-dimensional interpolation method before and after compensation is shown in Figure 7. Comparing Figures 7(a) and (b), it can be seen that the reflection peak

broadens considerably when it is not compensated, and its interval is approximately 49.5~73.5 m. However, the reflection peak after compensation is extremely sharp, indicating that the spatial resolution of the system has been greatly improved.

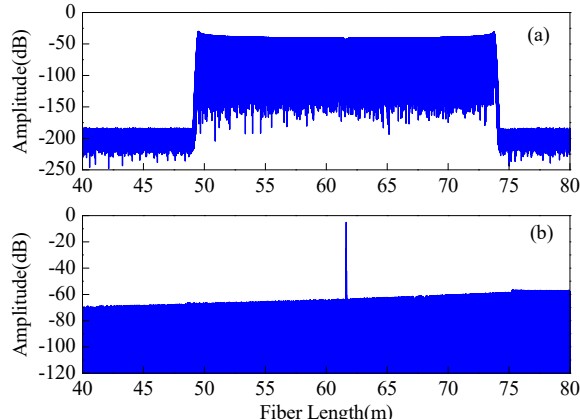

**Figure 7: The one-dimensional interpolation method to compensate for nonlinear single points, $A_n = 5 \times 10^6$, $f_n = 25$ Hz, and $\tau = 6 \times 10^{-7}$ s. (a) distance domain signal of the main interferometer without compensation; (b) distance domain signal of the main interferometer after compensation.**

(2) To test the compensation effect of the one-dimensional interpolation method for multiple reflection points, a second strong reflection peak position is added, namely $\tau_1 = 4 \times 10^{-7}$ s and $\tau_2 = 6 \times 10^{-7}$ s. At this time, the preset reflection point





positions are now at 41.10 m and 61.65 m. When uncompensated, as shown in Figure 8(a), the two reflection peaks overlap and the overlapping interval is observed at approximately 33.0–73.5 m. The position information of the two reflection points cannot be obtained at all and the system spatial resolution is seriously degraded. After compensation by the one-dimensional interpolation method, as shown in Figure 8 (b), the two reflection points are separate and the widths of their reflection peaks are significantly narrower. The corresponding fiber positions are also consistent with the preset positions. It is thus demonstrated that the cubic spline interpolation method effectively removes the phase noise in the light source and the compensation effect for multiple reflection points is still significant.

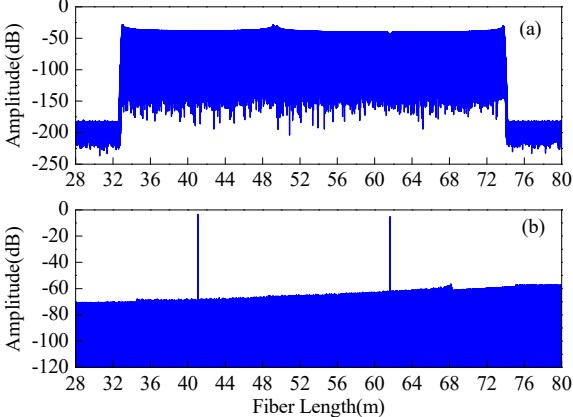

**Figure 8: One-dimensional interpolation method to compensate nonlinear double points, $A_n = 5 \times 10^6$, $f_n = 25$ Hz, $\tau_1 = 4 \times 10^{-7}$ s, and $\tau_2 = 6 \times 10^{-7}$ s (a) the distance domain signal of the main interferometer without compensation (b) the distance domain signal of the main interferometer after compensation.**

For the case of the same simulation model, in the face of more severe nonlinear frequency sweeping effects of the light source, the one-dimensional interpolation method greatly improves the spatial resolution of the system, narrows the width of the reflection peaks significantly, and obtains the position information of the reflection points. As with single-point compensation, one-dimensional interpolation is performed on the same set of experimental data. Figure 9 shows the effects before and after compensation.

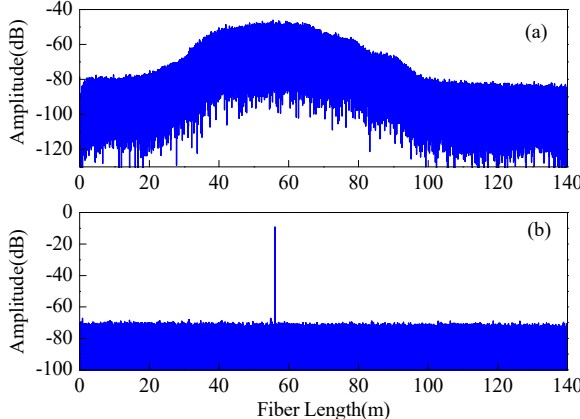

**Figure 9: The one-dimensional interpolation method compensates for the nonlinearity of the light source and the position of the reflection point of the test fiber is approximately 56 m. (a) distance domain signal of the main interferometer without compensation; (b) distance domain signal of the main interferometer after compensation.**

Figure 9(b) shows that after the cubic spline interpolation method, the problem of reflection peak expansion has been solved. There is an extremely sharp reflection peak at 56 m, the energy is concentrated, and the system spatial resolution is greatly



improved: the position information of the testing fiber reflection points is restored again. Therefore, the cubic spline interpolation method has notable compensation effects that can achieve higher system spatial resolutions and have lower noise floors and higher system signal-to-noise ratios.

**4 Calibration experiments**

    To measure and calibrate the lateral stress, the testing fiber consists of three sections with a total length of 809.29 m and the

lengths of the three segments are 595.96 m, 3.43 m, and 209.9 m; the fiber with the length 3.43 m has a plastic outer tube, and this section is mainly used for lateral stress application and measurement, while the remaining two sections consist of ordinary single-mode fiber.

    In the calibration experiment, the light source uses a Santec TSL-710 tunable wavelength scanning light source. The light source power is set to 10 mW, the frequency sweep rate is 16 nm/s, the acquisition card sampling rate is 250 MSa/s, and the

number of sampling points is 4 M. When no stress is applied, i.e., the elastic diaphragm is in a natural state and the nonlinearity of the light source is compensated by the dechirp declination filter algorithm, the distance signal obtained is shown in Figure 10. The figure shows that after compensation, the reflection peak is clearly visible and the spatial resolution is significantly improved. Simultaneously, the amplitude of the Rayleigh scattering spectrum is relatively uniform and does not exhibit major changes.

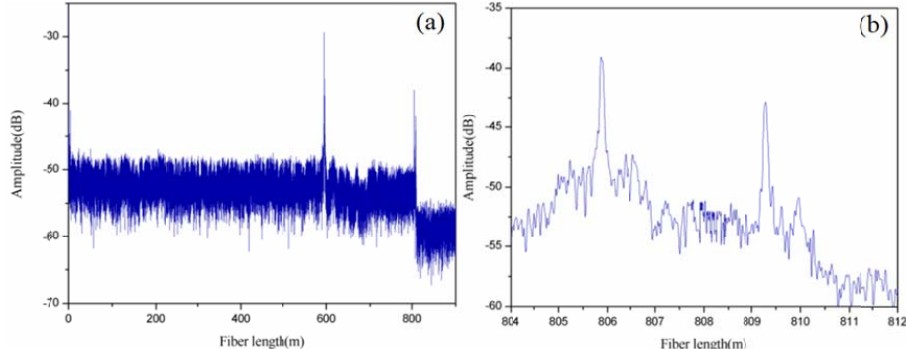


**Figure 10: Distance domain signal diagram of the test fiber without stress. (a) complete image for the distance domain; (b) enlarged image of the reflection points at 805.86 m and 809.29 m.**

    To measure the changes in the backscattered Rayleigh signal under different stresses, as described in Section 2.2, weights with different masses are applied to the sensor head; the weights are 5 kg, 10 kg, 15 kg, 20 kg, 30 kg, 54 kg, and 77.5 kg.

(The stress area of the compressing tablet in Figure 2 is $7.25 \times 10^{-4}$ m$^2$, providing corresponding pressure values of 0.069 Mpa, 0.138 Mpa, 0.207 Mpa, 0.276 Mpa, 0.414 Mpa, 0.746 Mpa, and 1.059 Mpa). The distance domain images of the backscattered signals under different lateral stress conditions are shown in Figure 11.

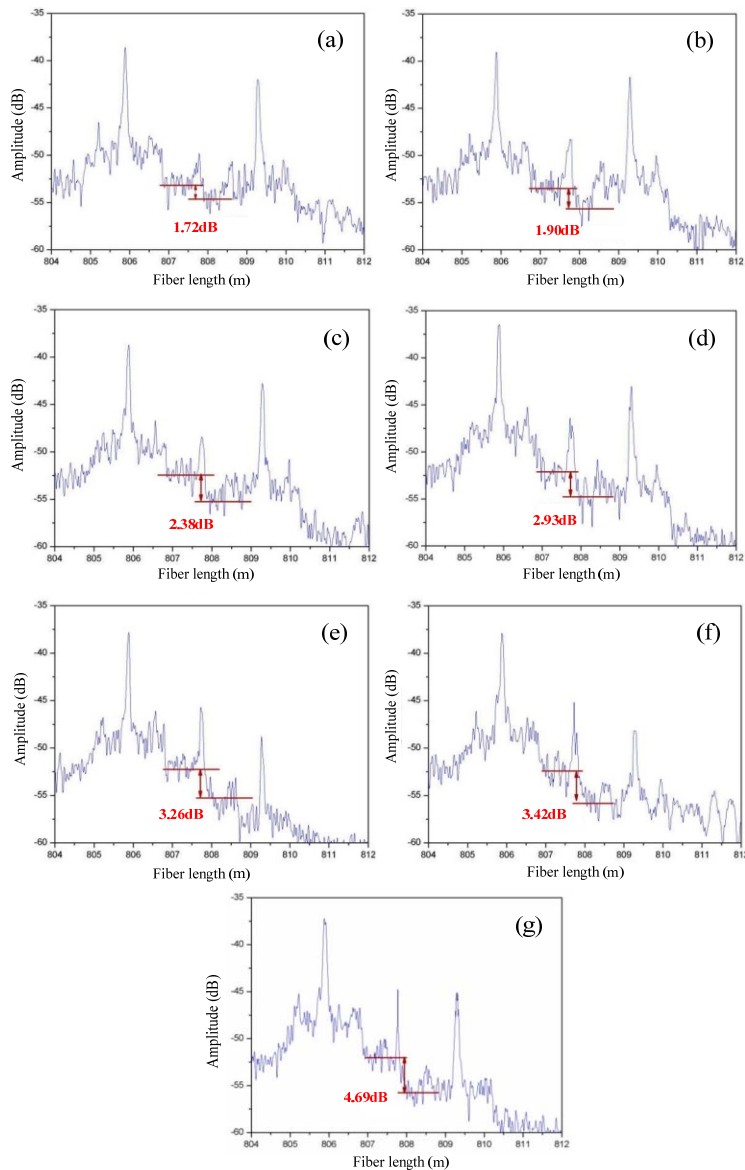

**Figure 11: Distance domain images of the OFDR signals under different stresses as follows: (a) 5 kg, (b) 10 kg, (c) 15 kg, (d) 20 kg, (e) 30 kg, (f) 54 kg, and (g) 77.5 kg.**


Due to the stress application, a reflection peak appears at this position, the loss difference can be obtained by averaging the amplitudes of the 100 points to the left and right of the reflection peak and subtracting them. The figure shows that with the continuous increase of the mass of the weight, the fiber is squeezed more by the pressing teeth, the larger the bending curvature radius generated, the more the Rayleigh scattering signal decreases. Simultaneously, greater stresses produce

higher peak values for the reflection peak caused by the bending, which reduces the optical power received by the subsequent optical fiber and causes the overall reflection peak generated at the end of the fiber to decrease in amplitude. It can also be seen from Figure 11(g) that when the mass of the weight reaches 77.5 kg, the amplitude of the Rayleigh scattering spectrum after the stress application point is still greater than the noise amplitude, indicating that the measurement range of the system has not reached its limit. As the mass of the weight continues to increase, the Rayleigh scattering



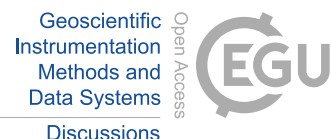

spectrum continues to decrease until it is flooded by noise in the fiber at the back of the sensor head and the peak-to-peak reflection amplitude at the end of the fiber is reduced to the noise amplitude. At this time the fiber is nearly broken.

**Table1.** Relationship table between lateral stresses and Rayleigh scattering signal intensity differences

| No. | Counterweight (kg) | Light intensity difference (dB) |
|-----|--------------------|----------------------------------|
| 1 | 5 | 1.72 |
| 2 | 10 | 1.9 |
| 3 | 15 | 2.38 |
| 4 | 20 | 2.93 |
| 5 | 30 | 3.26 |
| 6 | 54 | 3.42 |
| 7 | 77.5 | 4.69 |

The measured weight masses and the Rayleigh signal intensity difference data are listed in the Table 1 and polynomial fitting
is performed. The fitting results show that the coefficients of the fourth-order polynomial and the higher-order terms are very small and can be ignored. Herein, a cubic polynomial is used for fitting and the fitting results are shown in Figure 12. The signal strength difference measured by using a value of 20 kg is 2.93 dB and the mass of the calculated weight is 21.78 kg. Thus, it is known that the measurement error of the stress is 8.9%.

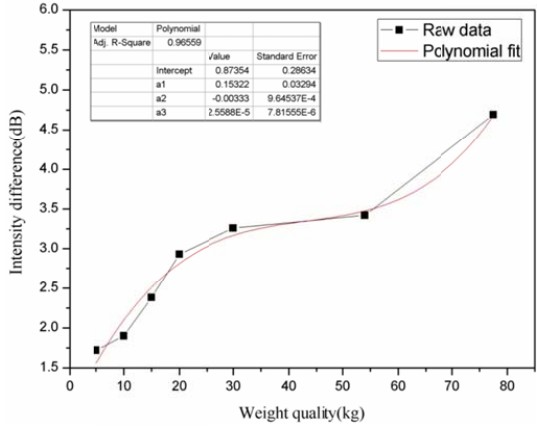

**Figure 12: Fitting graph of the weight masses and signal intensity differences.**

## 5 Field Engineering Application

### 5.1 Description of the Field Engineering Application Site

To verify the effectiveness of the cubic spline interpolation method in compensating for the nonlinear frequency sweep effect of the light source, this section uses LabVIEW software for simulation and then compares this simulation with the
laboratory experimental data.

The Chenjiagou landslide group is located on the slope of the left bank of the Meixi River, Yeoshe Village, Yaodi Village, Baidi Town, Fengjie County, and Chongqing City, located approximately 1 km from Fengjie Old County and 0.4 km from the Yangtze River. There are two large gullies on both sides of the main sliding body. The main sliding body boundary for which the right boundary is the Yaowancun Gully is oriented NE–SW; the left boundary is Chenjiagou and the right





boundary is approximately 340 m away from the NE point of the Fujun Bridge; the trailing edge has an elevation of approximately 385 m and a width of approximately 100 m; the leading edge has an elevation of 135 m and this side of the Linmeixi River is submerged. The main axial length of the landslide is approximately 500 m and its width is approximately 380–400 m, the average thickness of the landslide is 60 m, and the thickest part is approximately 99.35 m in the middle of the landslide. The existing distribution area of the landslide is approximately $18.4 \times 104$ m2, its volume is approximately

$1.104 \times 107$ m3, and it is a second-class large-scale soil landslide (Wu et al., 2005). Figure 13(a) shows the entire Chenjiagou landslide.

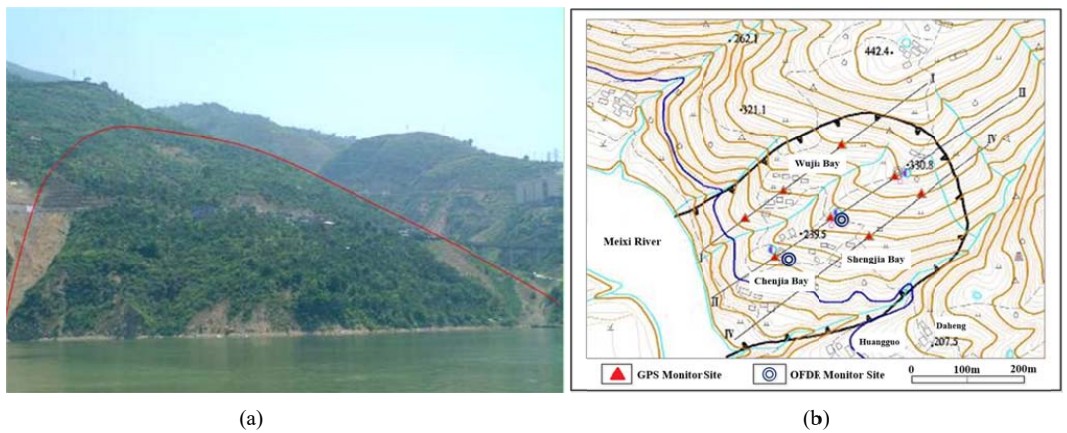

(a)            (b)

**Figure 13: The Chenjiagou landslide. (a) overview of the landslide; (b) schematic of the monitoring points.**

According to the characteristics of the Chenjiagou landslide and its site survey, we mainly conducted GPS surface displacement monitoring and internal thrust monitoring of the landslide. The locations of the monitoring points are shown in Figure 13(b). For the inner thrust monitoring scheme of the landslide body, we used an OFDR-based quasi-distributed optical fiber stress sensing system and installed it in the annular gap between the thrust measurement tube and the borehole, as shown in Figure 14. By measuring the pressure on each pressure sensor in the four directions of the design installation

hole depth (segmented by the sliding zone), the thrust force of each hole segment is obtained by the segmented integral method to determine the force of the slip zone, creating conditions that can provide reliable data support for the entire landslide monitoring process.

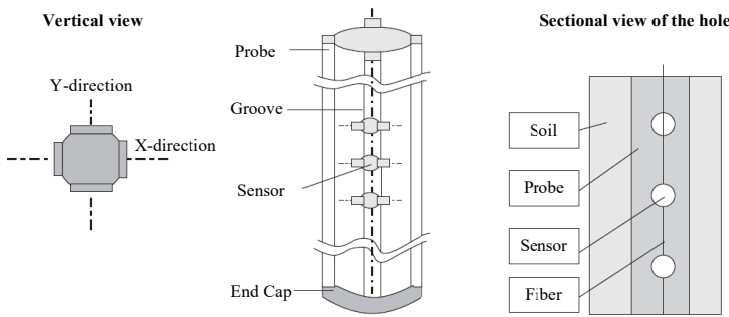

**Figure 14: OFDR sensor installation method.**

**5.2 Data Analysis**

As shown in Figure 13(b), the main section of the Chenjiagou landslide is equipped with two optical fiber thrust monitoring holes. Therefore, in this field application, the OFDR prototype was used to perform inner thrust tests of the two monitoring holes in the sliding body. Since the OFDR demodulator cabinet is placed at a distance from the thrust monitoring hole on the





landslide body, a fiber optic jumper with a length of 220 m is used to connect the cabinet to the fiber connector in the

borehole. Photos of the Chenjiagou landslide monitoring sites and the data collection equipment are shown in Figure 15.

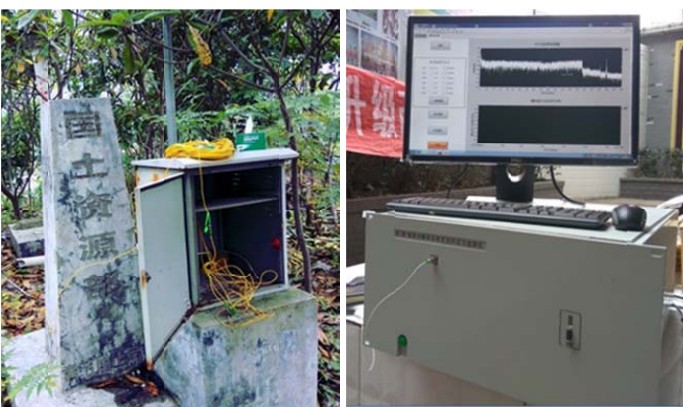

**Figure 15: Photos of Chenjiagou landslide monitoring site and the data collection equipment.**

### 5.2.1 Data analysis of the TK-01 monitoring site

Figure 16 shows the OFDR signal measured by the TK-01 fiber that was connected to the fiber jumper. The start and stop

positions of the TK-01 testing fiber are 222.849 m and 424.075 m, respectively, and the length of the testing fiber is 202.226 m. The positions of the OFDR sensors and the pressure values subjected to the lateral thrust can be obtained by processing the data of the OFDR signals, as shown in Figure 17.

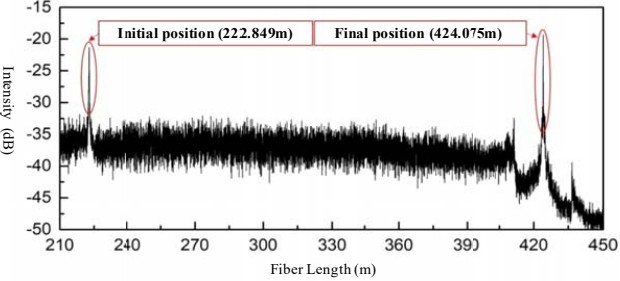

**Figure 16: OFDR signal of the TK-01 testing fiber.**

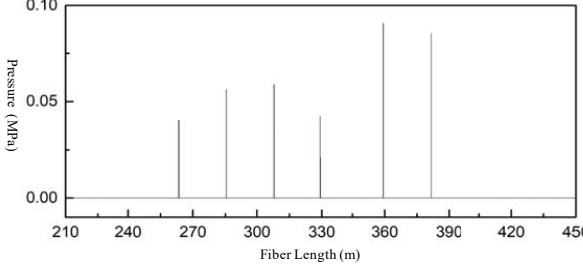


**Figure 17: Pressure positions and their values for the TK-01 testing fiber.**

Figures 16 and 17 show that the testing fiber exhibits obvious signal strength differences at the six positions that are 261.752 m, 286.213 m, 308.910 m, 329.585 m, 359.518 m, and 381.501 m (including the 220 m optical fiber jumpers). There are obvious signal intensity differences at these locations that are caused by the stresses on the sensors. The signal strength

difference showing a strong reflection peak at 425.075 m is caused by Fresnel reflection at the end of the sensing fiber.





The test results of the stress positions show good agreement with the positions of the sensors installed along the sensing fiber (the actual installation positions of the sensor are 40 m, 65 m, 90 m, 110 m, 140 m, and 160 m). Table 2 shows that there are six stress sensors along the testing fiber, and the measured positions and pressure values for each sensor are shown in Table 2.


**Table2.** Positions and measured pressure values of sensors of TL-01

| No. | Installation position (m) | Measured position (m) | Location error (m) | Measured pressure (MPa) |
|---|---|---|---|---|
| 1 | 40 | 41.752 | 1.752 | 0.04026 |
| 2 | 65 | 66.213 | 1.213 | 0.05627 |
| 3 | 90 | 88.910 | 1.090 | 0.0589 |
| 4 | 110 | 109.585 | 0.425 | 0.04215 |
| 5 | 140 | 139.518 | 0.498 | 0.09072 |
| 6 | 160 | 161.501 | 1.501 | 0.08539 |

**5.2.2 Data analysis of the TK-02 monitoring site**

Figure 18 shows the OFDR signal measured by the TK-02 fiber that is connected to the fiber jumper. The start and stop positions of the testing fiber are 222.053 m and 421.506 m, respectively, and the length of the testing fiber is 199.453 m. The positions of the OFDR sensors and the pressure values subjected to lateral thrust can be obtained by processing the data of

the OFDR signals, as shown in Figure 19.

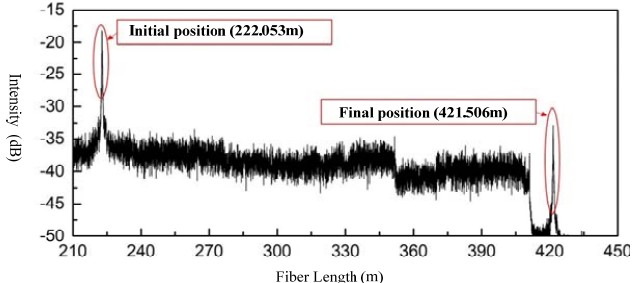

Figure 18: OFDR signal of the TK-02 testing fiber.

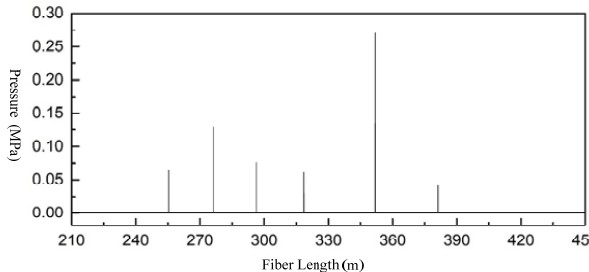

Figure 19: Pressure positions and their values for the TK-02 testing fiber.

Figures 18 and 19 show that the testing fiber exhibits obvious signal strength differences at the six positions that are 255.952 m, 276.255 m, 296.933 m, 322.125 m, 352.520 m, and 378.568 m (including the 220 m optical fiber jumpers). There are obvious signal intensity differences at these locations and these are the differences caused by the stresses on the sensors. The signal strength difference with a strong reflection peak at 425.075 m is caused by Fresnel reflection at the end of the sensing fiber.



The test results of the stress positions agree well with the sensor positions installed on the sensing fiber (the actual installation positions of the sensors are 35 m, 55 m, 75 m, 100 m, 130 m, and 170 m). Table 2 shows that there are six stress sensors on the testing fiber, and the positions and measured pressure values of each sensor are also shown in Table 3.

**Table3.** Positions and measured pressure values of sensors of TL-02

| No. | Installation position (m) | Measured position (m) | Location error (m) | Measured pressure (MPa) |
|-----|---------------------------|-----------------------|--------------------|--------------------------|
| 1 | 35 | 35.952 | 0.952 | 0.06505 |
| 2 | 55 | 56.255 | 1.255 | 0.13003 |
| 3 | 75 | 76.933 | 1.933 | 0.07549 |
| 4 | 100 | 102.125 | 2.125 | 0.06099 |
| 5 | 130 | 132.520 | 2.520 | 0.27088 |
| 6 | 160 | 158.568 | 1.432 | 0.0412 |

**6 Discussion**

The OFDR beat frequency is linearly proportional to the time delay ($\tau$). The beat point frequency can be used to locate the measurement point with high accuracy. Therefore, OFDR has high requirements for the frequency linearity of the sweep frequency source. The optical frequency of the actual sweep frequency light source changes nonlinearly with time. At present, software methods are mainly used to compensate for the nonlinear effect of the light source such as the non-uniform fast Fourier transform method (NUFFT), the one-dimensional interpolation method, and the dechirp filter algorithm. The

compensation process is shown in the Figure 20.

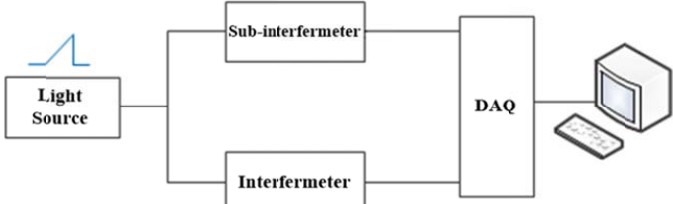

**Figure 20: Software methods to compensate light source nonlinearity.**

The NUFFT method transforms the data from a non-uniform grid to a uniform grid (Duijndam and Schonewille, 1999). The primary function of the NUFFT method is to compensate for the nonlinearity of the light source and to obtain instantaneous

optical frequency information of the sweep frequency light source from the sub-interferometer. The specific process is to obtain the beat frequency signal with time delay information from the sub-interferometer and to obtain the instantaneous optical frequency $v(t)$ after use of a Hilbert transform to normalize and discretize the instantaneous optical frequency to obtain $v_n$. The NUFFT method is then used to calculate the distance domain information. This method mainly uses the convolutional property of the Fourier transform (e.g., the Fourier transform of the convolution of two functions is equal to

the product of the respective Fourier transforms of the two functions) to perform the deconvolution operation, as shown in Equation (2) (Fessler, 2007).

$$X(z_n) = \frac{X_G(z_n)}{W(z_n)} \qquad (2)$$

In Equation (2), $W(z_n)$ is the Fourier transform of a Gaussian function $w(v_n)$. Similarly, LabVIEW software is used to simulate Equation (1) to verify the compensating effect of the NUFFT method. Parameter setting is the same as that for the


one-dimensional interpolation method discussed in Section 3. During the simulation process, $I(t)$ needs to be discretized while considering the size of the nonlinear effect.

(1) When the short-distance measurement is in the range 0–40 m, the nonlinear frequency sweep effect of the light source becomes larger without compensation as shown in Figure 21(a). When the NUFFT method is used for compensation, Figure 21(b) shows that broadening phenomenon of the reflection peak owing to the nonlinear effect is greatly improved and there

is a significant reflection peak at 20.55 m. Therefore, under short-range measurement conditions, the NUFFT method can effectively compensate for the nonlinear frequency sweep effect of the light source and improve the spatial resolution of the system.

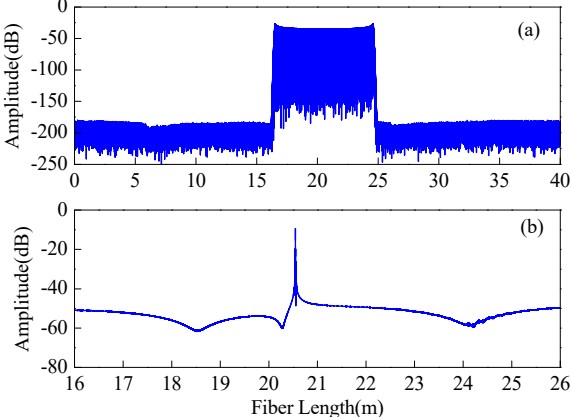

**Figure 21: NUFFT method to compensate for the nonlinearity of the light source under short-range measurement**
**conditions: $A_n = 5 \times 10^6$, $f_n = 25$ Hz, and $\tau = 4 \times 10^{-7}$ s. (a) distance domain signal of the main interferometer without compensation; (b) distance domain signal of the main interferometer after compensation.**

(2) When the measurement distance increases to 80 m, a comparison of Figure 21(a) and Figure 22(a) shows that the width of the broadened reflection peak (with no compensation) further increases, indicating that when the measurement distance increases, the linear effect grows stronger. Figure 22(b) shows the result after NUFFT compensation. It can be seen that

NUFFT effectively eliminates the phase noise of the light source and can clearly distinguish a strong reflection point around 61.65 m.

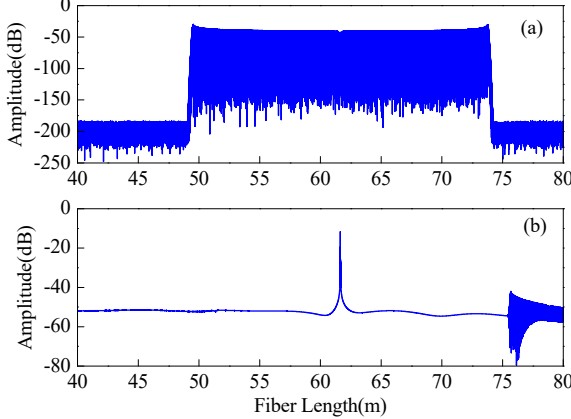

**Figure 22: NUFFT method to compensate for light source nonlinearity under longer-distance conditions (a) distance domain signal of the main interferometer without compensation (b) distance domain signal of the main**
**interferometer after compensation.**





However, Figure 22(b) shows that the light source signal exhibits drastically changing noise after 76 m that may indicate the need for deconvolution when performing NUFFT operations. As shown in Figure 23, the Gaussian function spectrum $W(z_n)$ is the dividend in Equation (2) that shows a good concentration. Therefore, its value is small and is close to zero at long distances (e.g., after 76 m) and the divisor $X_G(z_n)$ in Equation (2) is also close to zero at this distance. When two extremely

small numbers are divided, this operation may cause major errors and even exceed the actual nonlinear phase noise.

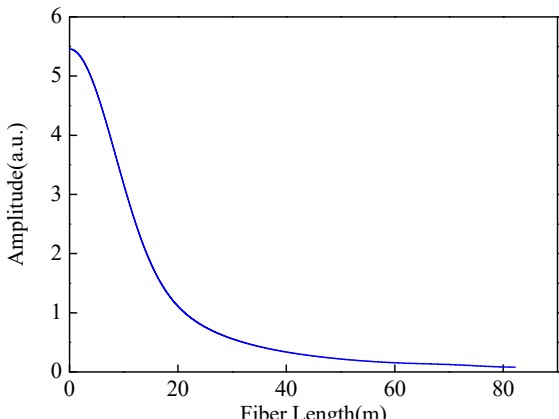

**Figure 23: The form of the Gaussian function distance domain in the NUFFT.**

Similarly, the NUFFT method is used to compensate for the single and double reflection points. From the simulation results, we know that within a short measurement range (generally no greater than 40 m), the main interferometer shows a narrow

reflection peak in the distance domain signal, which means the system spatial resolution has greatly improved. However, when the measurement distance exceeds 40 m, two minimal vectors, $W(z_n)$ and $X_G(z_n)$, will occur for division when the NUFFT is deconvolved. There will be side-lobes around the reflection peak indicating that the phase noise has not been completely eliminated; therefore, using the NUFFT method for light source compensation will generate larger errors and risks in the long-distance measurements.

Both the NUFFT and one-dimensional interpolation algorithms are resampling methods. These methods can achieve high spatial resolution for measurements over short distances. However, when the test distance increases, the difference between the test point delay on the main interferometer and the auxiliary interferometer delay becomes too large. Compared with the dechirp filter algorithm, the compensation effect is not as effective. When using the dechirp filter algorithm, the spatial resolution cannot be as high as when using resampling at short distances. Therefore, a compensation method combining a

resampling method and de-slope filtering algorithm should be studied in the future work to improve the nonlinear compensation effects for the OFDR light source that can achieve higher spatial resolutions for both short and long-distance measurements.

**7 Conclusions**

This paper proposes a quasi-distributed thrust measurement system based on OFDR. By designing an optical fiber stress

sensor based on the optical micro-bending effect, combined with a high-resolution and high-precision OFDR demodulator, and after a field engineering application on a landslide in the Three Gorges Dam area, quasi-distributed stress monitoring is accomplished with long measurement distances, high spatial resolutions, high sensitivities, and rapid responses. This paper provides a new concept and method for inner stress testing of rock and soil masses and can be extended to such safety-related monitoring fields as unstable slopes, slope engineering, water conservation and hydropower dams, and tunnel

chambers that have great practical value and application prospects.





The highlights of this paper are as follows:

(1) Cubic spline interpolation is used to improve the spatial resolution of the OFDR sensor components. This algorithm compensates for the nonlinear effects of the light source, improves the resolution and spatial-positioning capability of the sensing system, and also improves the signal-to-noise ratio of the system. LabVIEW simulations and laboratory tests show

that the OFDR sensing system achieves a spatial resolution of 20 cm in a 500 m testing fiber.

(2) The OFDR sensing system is calibrated by conducting a lateral stress laboratory experiment. The maximum measurement pressure reaches 1.059 MPa and the maximum relative error is 8.9%. The field engineering application for the Chenjiagou landslide shows that the system can accurately locate multiple fiber micro-bending stress sensors installed in the landslide body within a range of 0–420 m and can obtain the pressure values of the lateral thrust.


**Author contributions.** Yimin Liu and Pu Wang designed and built the instrument with the help of Chenghu Wang. Yimin Liu and Chenghu Wang prepared the paper with contributions from all authors.

**Competing interests.** The author declares that there is no conflict of interest.

**Acknowledgements.** Ce Zhou and Wenjun Chen helped acquire the data of engineering application presented in this paper.

**Financial support.** This research has been supported by the National Natural Science Foundation of China, grant no. 41804089; National Natural Science Foundation of China, grant no. 61327004.

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
