# Peer review of "Research and Application of an Inner Thrust Measurement System for Rock and Soil Masses based On OFDR"

_Geoscientific Instrumentation, Methods and Data Systems, 2020_

## Referee Comment (RC1) · Anonymous Referee #1 · 13 May 2020

This manuscript has some language and technical issues: 1. Line 24: "prospects" is wrong. 2. Line 28: "domestically and abroad" is wrong. 3. Line 35: "the disaster body movement" is wrong. 4. Line 45: "Dehua Liu et al." is wrong format of citation. 5. Line 58: "Bin Shi et al." is wrong format of citation. 6. Line 67: Should "demodulator" be interrogator? 7. Line 226: "Mpa" should be "MPa". 8. Figure 14 is very confusing. A 3D cartoon is required. 9. How does the temperature affect the measurement? 10. For monitoring purpose, time history of the pressure is desirable. However only data points are listed in Table 3.
* * *
[Figure]

https://doi.org/10.5194/gi-2020-6, 2020.

---

## Author Comment (AC1) · 19 May 2020

Dear reviewer 1#ïïjŽ

We are very grateful to this referee comments, and we have carefully read and considered the referee's comments, and these comments are important for improving the quality of this manuscript. Based on these comments, we have made carefully modification of language and technical issues in the original manuscript, the detail modifications are shown in next chapter.

Thank you very much for your suggestion and consideration, and we look forward to

hearing from you. Best regards,

Yimin Liu and Chenghu Wang.

Detailed revision:

(1) Line 24: "prospects" is wrong.

Modification: In the previous manuscript, "has good engineering application prospects in the safety monitoring field", it is not so accurate and clear. We revise it as "has a good application prospect in the safety monitoring field".

(2) Line 28: "domestically and abroad" is wrong.

Modification: We revise it as "at home and abroad".

(3) Line 35: "the disaster body movement " is wrong.

Modification: We revise it as "early warning and prediction of the geological disaster cannot be accurately achieved".

(4) Line 45: "Dehua Liu et al." is wrong format of citation.

Modification: We revise it as "Liu et al.".

(5) Line 58: "Bin Shi et al." is wrong format of citation.

Modification: We revise it as "Shi et al.".

(6) Line 67: Should "demodulator" be interrogator?

Explanation: The fiber demodulator is the device that converts optical signals into electrical signals, so this place should be "demodulator".

(7) Line 226: "Mpa" should be "MPa".

Modification: We are sorry for this error, and correct them.

(8) Figure 14 is very confusing. A 3D cartoon is required.

Modification: We have redraw a 3D figure for the OFDR sensors installation.

(9) How does the temperature affect the measurement?

Explanation: Different from fiber Bragg grating, the polarization-preserving single-mode fiber used in this OFDR sensing component is not affected by temperature. Moreover, the sensor assembly is installed in the deep rock mass, and generally does not consider the error caused by temperature change.

(10) For monitoring purpose, time history of the pressure is desirable.

Modification and explanation: Tables 3 and 4 show positioning accuracy of the OFDR sensing component and the pressure value in the first measurement. And your suggestion is reasonable and necessary, we have selected the pressure monitoring data of TK-01 monitoring site from February 2017 to December 2019, and the data table and pressure-time curve are shown in table 1 and figure 2. These data and the curves show that, this OFDR thrust measurement system operates normally for a long time, and the measured data of OFDR sensor component are relatively stable, which means the landslide is in a creep state. And we have added this content from line 306 to 315.
* * *
[Figure]

Figure 1: OFDR sensors installation method

**Fig. 1.**

Table 1. The pressure data table of TK-01

| Measurement Time | 1# Sensor | 2# Sensor | 3# Sensor | 4# Sensor | 5# Sensor | 6# Sensor |
| --- | --- | --- | --- | --- | --- | --- |
| 2017/2/1 | 0.0403 | 0.0563 | 0.0589 | 0.0422 | 0.0907 | 0.0854 |
| 2017/5/5 | 0.0453 | 0.0653 | 0.0609 | 0.0405 | 0.0885 | 0.0863 |
| 2017/8/15 | 0.0545 | 0.0764 | 0.0779 | 0.0400 | 0.1154 | 0.0854 |
| 2017/9/14 | 0.0519 | 0.0763 | 0.0754 | 0.0325 | 0.1134 | 0.0868 |
| 2017/11/15 | 0.0462 | 0.0662 | 0.0765 | 0.0399 | 0.1200 | 0.0862 |
| 2018/2/13 | 0.0457 | 0.0668 | 0.0741 | 0.0302 | 0.1053 | 0.0854 |
| 2018/5/14 | 0.0562 | 0.0685 | 0.0699 | 0.0266 | 0.1193 | 0.0863 |
| 2018/7/10 | 0.0554 | 0.0713 | 0.0706 | 0.0321 | 0.1193 | 0.0860 |
| 2018/8/17 | 0.0662 | 0.0746 | 0.0724 | 0.0303 | 0.1104 | 0.0862 |
| 2018/9/11 | 0.0625 | 0.0710 | 0.0703 | 0.0265 | 0.1255 | 0.0855 |
| 2018/12/19 | 0.0557 | 0.0700 | 0.0685 | 0.0325 | 0.1188 | 0.0867 |
| 2019/2/16 | 0.0512 | 0.0751 | 0.0710 | 0.0299 | 0.1104 | 0.0888 |
| 2019/5/15 | 0.0572 | 0.0891 | 0.0700 | 0.0365 | 0.1223 | 0.0862 |
| 2019/7/17 | 0.0592 | 0.0769 | 0.0751 | 0.0333 | 0.1224 | 0.0867 |
| 2019/8/16 | 0.0685 | 0.0899 | 0.0891 | 0.0370 | 0.1235 | 0.0867 |
| 2019/12/24 | 0.0652 | 0.0799 | 0.0875 | 0.0370 | 0.1158 | 0.0863 |

**Fig. 2.**

[Figure]

Figure 2: Monitoring data curves of multiple sensors

**Fig. 3.**

---

## Referee Comment (RC2) · Anonymous Referee #2 · 3 Jun 2020

The paper by Liu and Co-authors is about a quasi distributed system to measure "stress." First of all, I found the motivation not adequate: the Authors start by claiming that FBGs show low spatial resolution, low measurement accuracy, and non-distributed measurements, but the proposed solution can be compared to FBGs in term of accuracy, accuracy. About the distributed feature, neither the proposed solution is distributed. Then they claim generally that "the sensing fiber contains may pressure-sensing points in series", which is meaningless, without a proper context. About the SOTA: OTDR has been proposed in some marginal applications for distributed monitoring, but it is not feasible for monitoring for the reasons also enlisted by the Authors. Talking about BOTDR, compared to other techniques, such as BOTDA/BOFDA

or OFDR, it is well known that it does not provide high accuracy. About the capability of achieving simultaneous measurement of strain and temperature over the same fiber, to my knowledge, it has been demonstrated in BOTDA schemes, but it requires a complex setup to measure both the Brillouin shift and power spectrum (supported by Rayleigh measurement). Alternatively, special fibers (LEAF) should be used. Most typically, in BOTDR systems, one collects the shift, and it is not possible to distinguish among the temperature and strain effect. Then, the Authors claim that "Optical frequency domain reflection (OFDR) presents the advantages of quasi-distributed monitoring..." but it is not the case: OFDR is normally used as a distributed sensing system, despite the fact that the Authors used it as a quasi distributed system. The Authors ultimately designed a transducer that induces pressure-dependant losses and measure it using an OFDR. But, in my opinion, this is a reductive way to use a powerful tool such as an OFDR. Typically, an OFDR is used to measure the Rayleigh spectral shift induced by the strain and gives the best performance when used in such a way. When used as an intensity-based system, it is affected by large uncertainties.

About the cubic spline interpolation, it is a well known approach, introduced, for example in: Yüksel K., Wuilpart M., Mégret P. Analysis and suppression of nonlinear frequency modulation in an optical frequency-domain reflectometer. Opt. Express. 2009;17:5845–5851. doi: 10.1364/OE.17.005845. Vergnole S., Lévesque D., Lamouche G. Experimental validation of an optimized signal processing method to handle non-linearity in swept-source optical coherence tomography. Opt. Express. 2010;18:10446–10461. doi: 10.1364/OE.18.010446. Song J., Li W., Lu P., Xu Y., Chen L., Bao X. Long-Range High Spatial Resolution Distributed Temperature and Strain Sensing Based on Optical Frequency-Domain Reflectometry. IEEE Photonics J. 2014;6:6801408. doi: 10.1109/JPHOT.2014.2320742. Kim D.Y., Ji Y.L., Ahn T.J. Suppression of nonlinear frequency sweep in an optical frequency-domain reflectometer by use of Hilbert transformation. Appl. Opt. 2005;44:7630–7634.

The Authors should explain where is the novelty.

To conclude, I found the use of the OFDR technique to probe intensity-based sensors, not so indicated for this application, as the main feature of the technique cannot be exploited. The Authors claim that their system achieves a spatial resolution of 20 cm in a 500 m testing fiber. Still, I do not think that they will exploit such resolution in practice (it would mean having 2500 sensors). Given the requirements and experimental tests, in my opinion, a well-designed system made of arrays of FBG load cells may provide better performance at a lower cost.

---

## Author Comment (AC2) · 29 Jun 2020

Dear reviewer 2#:

We are very grateful to this referee comments, and we have carefully read and considered the referee's comments, and these comments are important for improving the quality of this manuscript.

We are very sorry that you are not satisfied with my article, but I need to explain some actual situations. You said that compares with the FBGs and BOTDR, the OFDR in my paper do not have much advantage, but we think the OFDR technology is very
suitable for to measure the inner stresses of rock and soil masses, which has a better spatial resolution and cost performance. With the micro-bending pressure sensor in section 2.1 and OFDR demodulator in section 2.2, the OFDR sensing system achieved a spatial resolution of 20 cm using a 500 m test fiber, and the resolution of an OTDR or BOTDR sensing system is generally approximately 0.8~1m. Compares with the FBGs, OFDR has no worse performance than FBG, and the field engineering application mentioned in section 5, over the years we have buried many micro-bending pressure sensors in the Three Gorges reservoir area in China, previously we used the OTDR method to measure the inner thrust, therefore, this study uses a more advanced OFDR method instead of OTDR, instead of using the FBGs. Therefore, the OFDR technology using in inner thrust measurement of rock and soil masses is necessary and reasonable.

Thank you very much for your suggestion and consideration, and we look forward to hearing from you.

Best regards, Yimin Liu and Pu Wang.